# Factors Informing the Development of a Clinical Pathway and Patients’ Quality of Life after a Non-Union Fracture of the Lower Limb

**DOI:** 10.3390/healthcare11121810

**Published:** 2023-06-20

**Authors:** Nontembiso Magida, Hellen Myezwa, Witness Mudzi

**Affiliations:** 1Department of Physiotherapy, Faculty of Health Sciences, University of Pretoria, Private Bag x323, Arcadia, Pretoria 0007, South Africa; 2Department of Physiotherapy, University of the Witwatersrand, Johannesburg 2193, South Africa; hellen.myezwa@wits.ac.za; 3Centre for Graduate Support, University of Free State, Bloemfontein 9301, South Africa; mudziw@ufs.ac.za

**Keywords:** clinical pathways, lower limbs, quality of life, non-union fractures

## Abstract

Patients with non-union fractures spend extended periods of time in the hospital following poor healing. Patients have to make several follow-up visits for medical and rehabilitation purposes. However, the clinical pathways and quality of life of these patients are unknown. This prospective study aimed to identify the clinical pathways (CPs) of 22 patients with lower-limb non-union fractures whilst determining their quality of life. Data were collected from hospital records from admission to discharge, utilizing a CP questionnaire. We used the same questionnaire to track patients’ follow-up frequency, involvement in activities of daily living, and final outcomes at six months. We used the Short Form-36 questionnaire to assess patients’ initial quality of life. The Kruskal–Wallis test compared the quality of life domains across different fracture sites. We examined CPs using medians and inter-quantile ranges. During the six-month follow-up period, 12 patients with lower-limb non-union fractures were readmitted. All of the patients had impairments, limited activity, and participation restrictions. Lower-limb fractures can have a substantial impact on emotional and physical health, and lower-limb non-union fractures may have an even greater effect on the emotional and physical health of patients, necessitating a more holistic approach to patient care.

## 1. Introduction

Complete knowledge of the clinical pathways (CPs) related to non-union fractures is imperative in order to guide healthcare practitioners in planning treatment and improving the quality of life of patients. Holistic patient care requires that quality of life should be managed and included in diverse CPs [1]. A multidisciplinary team of healthcare professionals should follow CPs that provide and manage all possible consequences for these patients to secure the best potential outcomes [2]. This aligns with the complex treatment and costs associated with managing non-union fractures [3,4], as well as the recurring follow-up required by patients to achieve positive outcomes.

Non-union fractures occur in about 2% of all fractures, but the prevalence of these fractures in diaphyseal fractures can be as high as 20% in certain situations [5]. Long bones are more prone to developing non-union fractures [5]. Non-union is defined as a permanent failure of the healing process, necessitating further treatment [6]. Non-union fractures may remain non-united for a period of six to nine months without healing or showing any signs of healing following injury [7]. The healing process of non-union fractures varies depending on anatomical region, pre-existing co-morbidities, and fracture type. In general, low-energy closed fractures are expected to heal significantly faster than high-energy open long-bone fractures due to the presence of soft tissue injuries [8]. Rather than relying on a certain time limit to describe non-union, it has been suggested that a more practical definition of non-union may be a fracture that will not heal without additional intervention [9].

A clinical route, or CP, is a multidisciplinary tool that standardizes treatment for a specific group of patients based on evidence-based practice [10]. CPs have been connected to more efficient systems by lowering in-hospital problems and errors, enhancing documentation, and encouraging favorable patient outcomes and safety [11,12]. Developing optimal CPs requires collaboration among all healthcare workers [13,14]. Developing CPs begins with selecting an appropriate clinical condition that has certain needs in terms of patient and clinical care [15]. CPs are evidence-based road navigators used to enhance overall patient outcomes [15]. CPs can also be improved and adapted to facilitate the monitoring of acute clinical care. Despite advancements in clinical care, the CPs for patients with non-union fractures of the lower limb remain unknown.

Non-union is a difficult outcome to predict at the time of injury and during the healing process [16]. A non-union fracture, like any other complication, is seen as an unsatisfactory outcome in clinical practice by doctors and patients [16]. These unfavorable outcomes may hamper patients’ quality of life. Many studies have reported a negative relationship between fracture healing and age, with non-union fractures associated with older age [3,17]. Non-union fractures also appear to be more common in patients with diabetes [18], HIV-related disorders [19], and present or past smokers [20,21].

In South Africa, many patients present with fractures and comorbidities that may impact fracture healing. Patients’ perspectives on their quality of life due to lower-limb non-union fractures have yet to be adequately recorded. With the reported prevalence of lower-limb non-union fractures [5], as well as the critical importance of lower-limb weight-bearing bones that carry the body, it is essential to determine the CPs of lower-limb non-union fractures. Poor clinical pathways may lead to a poor quality of life. This study aimed to determine CPs and assess the quality of life of patients with lower-limb non-union fractures in five Gauteng state hospitals in South Africa.

## 2. Materials and Methods

### 2.1. Study Design

A cross-sectional study design was used to collect baseline data while patients were still in the hospital. A prospective study design was used to collect data pertaining to their CPs, and we surveyed them to establish their quality of life.

### 2.2. Sample

Our sample included 22 patients diagnosed with lower-limb non-union fractures admitted to and treated in the orthopedic departments of five state hospitals in Gauteng, South Africa. Patients were recruited while still in the hospital and followed for a period of six months. Patients diagnosed with non-union lower-limb fractures were recruited consecutively to participate in the study, and those who did not reside in the Gauteng province were excluded for follow-up reasons. We also excluded patients with pathological fractures and patients with multiple fractures. The present study was approved by the Ethics Committee at the University of the Witwatersrand (M 150236) and the University of Pretoria (349/2017).

### 2.3. Methods

We reviewed admission registers, and all the patients with non-union lower-limb fractures in the various orthopedic wards of the different hospitals were recruited to participate in the study and asked to complete the Short Form-36 (SF-36) questionnaire. The questionnaire includes questions in eight domains that fall under the physical component score (PCS) and the mental component score (MCS). The PCS includes physical functioning, physical role, body pain, and general health; and the MCS evaluates mental health, vitality, emotional role, and social functioning. In addition to the SF-36 questionnaire (Appendix A), the CP questionnaire was used to collect data from admission until discharge. Data were collected from medical records and included a structured multidisciplinary team intervention. The information required for the CP questionnaire included the admission date, length of stay in the hospital, medication given in the hospital, postoperative complications, mobility status at discharge, return to work, return to previous home status, readmission within six months, health-related quality of life at home, and frequency of follow-up appointments.

### 2.4. Statistical Analysis

Data were collected and analyzed using Stata version 14. Data were presented as frequencies and percentages, and the corresponding 95% confidence interval was also provided for categorical measurements. In addition, the Kruskal–Wallis test, the non-parametric equivalent of the one-way analysis of variance (ANOVA), was used to compare two or more independent samples of equal or different sample sizes. In this study, we used the Kruskal–Wallis test to compare patients with different diagnoses within each domain of the SF-36, and data normality was tested using the Shapiro–Wilk test. The data were analyzed to assess the quality of life of patients with lower-limb non-union fractures. The Kruskal–Wallis test was performed with the hypothesis that several patients were from the same population.

## 3. Results

Over a period of six months, we prospectively surveyed patients with lower-limb non-union fractures in five state hospitals in Gauteng, South Africa. A total of 22 patients (91% male; mean age 40.68 ± 13.74 years) completed the CP and SF-36 questionnaire. The results are presented in Table 1.

Most patients were men, single, and younger than 60 years. Other socio-demographic characteristics of this group were as follows: 68% (*n* = 15) unemployed, 86% (*n* = 19) smokers, and 36% (*n* = 8) HIV positive. The distribution of non-union fractures in lower limbs, and the presentation domains, that is, impairment, activity, and the participation of patients, are presented in Table 2.

Most patients with lower-limb non-union fractures had fractures in the distal femur and distal tibia at 22.7% (*n* = 5) each. A total of 54.6% (*n* = 12) of the patients were readmitted to the hospital within a period of six months due to complications related to their non-union fractures. The length of hospital stay was a median of four months (IQR = 3–5).

### Quality of Life Measure

Data from the SF-36 were analyzed using the non-parametric Kruskal–Wallis Chi-squared test in order to assess the significant differences between the eight domains of SF-36 and the non-union fracture diagnosis. The Kruskal–Wallis Chi-squared test was selected because it is non-parametric, and the study had a small sample size. Table 3 reveals the results of the lower-limb non-union fracture diagnosis in relation to the SF-36 domains.

The results of the Kruskal–Wallis test demonstrated that the emotional role scores (*p*-value = 0.03) were significantly affected by the location of lower-limb non-union fractures. The physical health of patients was significantly (*p*-value = 0.0001) affected by the location of non-union lower-limb fractures. However, due to the study’s small sample size, these results should be interpreted cautiously as there were no adjustments made to the variables.

The results regarding the quality of life of patients with non-union fractures are presented in Table 4, according to the Physical Component Summary (PCS) score and the Mental Component Summary (MCS) score.

In general, all domains scored very low for quality of life in these patients. Both in the PCS and MCS domains, physical role and emotional role, respectively, scored very low in all patients with non-union fractures, indicating a poor quality of life. With all the poor quality of life demonstrated, patients with non-union fractures did their best to keep their general health (17; 16, 17) in all fractures assessed. Comparable to MCS, the mental health of all patients with non-union fractures was beheld, even though it was still low.

## 4. Discussion

In this study, we found that most patients with non-union fractures were younger than 60 years of age. A cohort study in the United States of America confirmed this decrease in non-union fracture with increasing age, reporting 2.5 times more non-union fractures in patients aged 30 to 44 years [22]. Another epidemiological study additionally suggested that individual counselling and treatment modification in this young group at-risk patients with non-union fractures be utilized [16]. It is crucial to include psychologists in the development of clinical pathways to support counseling. In contrast, previous studies have reported that non-union fracture complications are more prevalent with increasing age, as suggested by physiological changes, and are therefore susceptible to delayed healing [16]. One of these physiological changes is the disruption of an essential step in proper fracture healing, namely the inflammatory response, which could have a detrimental effect on the healing process [23]. Although our results may disqualify the ageing process as a major contributing factor in lower-limb non-union fractures, our results suggest that other factors, such as smoking (86%), HIV (58%), diabetes (86%) and other social influences, may play a role in non-union fracture prevalence. Nandra et al. supported and gave insight into bone density deficit in smokers [16], which is associated with delayed fracture healing and non-union. In modern times, patients use e-cigarettes, which are not making adverse events in fracture healing better [24]. In addition to smoking, HIV was another contributing factor noted as detrimental to non-union fracture healing. HIV may hinder fracture healing by reducing the bone mineral density, mineralization, and turnover. In a study investigating fracture healing in HIV-positive patients in South Africa, they found that 6% developed non-union after nine months [19]. Nevertheless, the rate of non-union fractures in HIV-positive patients is not widely different from non-HIV-positive patients, as some patients have multiple comorbidities like diabetes. However, we were unable to determine whether non-union fractures were prevalent because of HIV infection, as we included all patients regardless of their HIV status or any other comorbidities.

In our study of CPs, we included patients with lower-limb non-union fractures who were admitted to the orthopedic wards of various hospitals. We excluded patients not living in the Gauteng province and those with pathological and multiple fractures. A previous prospective study of femoral neck fractures in the elderly also excluded multiple fractures and fractures caused by malignancy [25]. Another study in Germany also excluded patients with cancer in their description of CPs for fractures of the pelvic ring [26]. Ref. [27] also excluded patients with pathological fractures, though the reason for exclusion was not stated. The inclusion and exclusion criteria in developing CPs are important because patients who are not part of the same follow-up process should be excluded when determining patient outcomes. Patients with malignancy-associated fractures may require extended treatment, and may have limited healing rates due to poor immunity associated with their condition. Patients with cancer may also need to be followed up more frequently, which would change their CP. Additionally, patients with multiple fractures may require different simultaneous treatments, which may mask the effect of particular CPs [28]. Compliance should always be measured prior to CP development and implementation in order to determine the impact on clinical practice. In this study, we investigated CPs by reviewing patients’ records in order to identify their date of lower-limb fracture, anatomical area of non-union, the medication used, and other healthcare practitioners involved. Identifying the healthcare providers, their working groups, and the education of their staff has been emphasized as a core implementation method [29]. Clinical pathways have the potential to lower treatment errors and improve patient outcomes by standardizing the clinical processes of care within the distinct culture and environment of the healthcare institution [30]. There is evidence that clinical medical records may improve the quality of care [31]. The author detailed the need for reliable information sources, solid interpersonal connections, and access to multidisciplinary teams of professionals and other specialists to properly address the multifaceted needs of patients [31]. Good clinical record-keeping promotes communication between various healthcare professionals and enables the continuity of care [32]. Included in the healthcare practitioners involved, physiotherapy services are essential. Although physiotherapy may not be accountable for promoting fracture healing, suitable therapeutic intervention may be responsible for eliminating complications that unreasonably slow fracture healing. Due to pain compensation, patients tend to maintain their limbs in shortened positions, reducing muscle length. This abnormal positional correction can ultimately lead to reduced truncal height [33]. Thus, physiotherapy aims to maintain muscle length using active physiological movements and maintain good posture in order to prevent bodily complications [34]. Posture correction aids in the prevention of secondary complications, such as spinal scoliosis, which may lead to a poor quality of life. Patients who develop long-bone non-union fractures have been reported to be at greater risk of emotional distress and poor physical health [35]. Limited studies were found to detail the use of physiotherapy treatment in post non-union fractures.

Other studies have retrospectively compared clinical outcomes before and after CPs were implemented [29,30]. The patient’s length of stay in our study was a median of four months (IQR = 3–5) in a follow-up period of six months, which is in contrast to a study conducted in Hong Kong, which only assessed the pre-operative length of stay [31]. The former study did not indicate the length of stay post-operatively. In contrast, the median length of hospital stays in a study conducted in Australia was only two days, with a median follow-up of eight months for fracture complications [3]. In African countries, social issues and wound infection may contribute to the length of hospital stay [32]. Given how social issues affect the length of hospital stay for non-union fractures, social workers must be included in clinical pathways. Liaison between patients’ families and healthcare professionals has been found to be an important task in facilitating smooth discharge from the hospital [33]. Additionally, nursing staff with expertise in wound care should be involved in the development of clinical pathways.

In our study, we assessed impairment, activity limitation, and participation restriction in activities of daily living. Patients’ final outcomes were also observed regarding whether they were healed or had returned to work following a lower-limb non-union fracture. We could not find any studies that reported the international classification of function (ICF) in non-union fracture CPs. The implementation of CPs may vary in lower-limb non-union fractures owing to different anatomical fracture sites and ICF scores due to different activities of daily living and socio-demographic characteristics. Lower-limb non-union fractures are linked to escalated morbidity, resulting in physical, mental, and emotional damage to the patient [34]. Our results show that the physical health and emotional domains of the patients were significantly affected by the location of lower-limb non-union fractures, though we cannot infer the findings as the sample size was too small. A similar study concurred that the outcomes for physical and psychological health were impacted in long-bone non-union fractures [35]. Regrettably, in a different South African study, some patients with non-union fractures expressed suicidal thoughts [36]. Suicidal expression highlights the necessity of involving psychologists in the development of clinical pathways. Our findings suggest that patients were still functioning physically, but their non-union fracture suppressed their mental health. This also affected their emotions even though their social functioning was not impeded. Another cross-sectional study in South Africa reasoned that declining physical health might also lead to poor psychological health [36], which was observed in the participants of our study.

Most of the patients in our study were men (*n* = 20), which precluded any comparisons between the sexes. The high proportion of male patients suggests that men may be more susceptible to non-union fractures in the South African context. This may be due to high smoking rates or highlight that men have higher levels of physical activity than women [37]. In the United Kingdom, more men than women were reported to have non-union fractures (57% vs. 43%) [38]. We found that most patients (86%) diagnosed with lower-limb non-union fractures were smokers. Fracture healing is a complex process in which biological, social, and systematic aspects interact and influence the healing time and possibility of non-union [39]. Smoking slows down healing as nicotine inhibits the replication of the different cells necessary for healing [40]. In a retrospective study in Australia, non-union fractures were the most common fracture healing complication compared to other fracture healing complications [3]. This study also found that 42% of non-union patients were either current or former smokers [3]. In our study, the patients were relatively young (40.68 ± 13.74 years) compared to previous studies [6]. Non-union fractures may be more problematic for younger adults and affect their quality of life to a greater degree since they cannot return to work. A study of patient risk factors in the United Kingdom reported that fracture complications were more prevalent in even younger patients (18–29 years) [35]. Young people may be more physically active and more prone to non-union and other fracture complications. Previous studies have usually reflected a negative correlation between fracture healing and age, and also an increase in the incidence of bone fracture with age [6]. Another study in California noted that advancing age reduces callus vascularization during fracture healing, thus likely decreasing oxygen levels at the fracture and limiting the exchange of other nutrients that are needed for healing [41].

We found that 86% of the patients who had non-union lower-limb fractures had diabetes. Studies have shown a strong link between diabetes and fracture healing complications, such as non-union [42]. Additionally, non-union fractures were diagnosed in 91% of the patients who were male and 58% of those who were HIV positive in this study. In South Africa, all patients with non-union fractures are screened for added comorbidities that are known to compromise fracture healing. HIV has been linked to bone loss in the past [43]. Whilst HIV-positive people have the same risk of fractures as HIV-negative people, they seem to experience between 30–70% more incidences of fractures due to low bone mineral density [44]. Another study noted a higher prevalence of fracture healing complications in HIV-positive male patients between the ages of 19 and 60 years [39,45]. In this study, we did not investigate how the bone that sustained the fracture is prone to non-union or the occurrence of a fracture. The investigators used the literature provided to quantify the fracture healing of non-union fractures. We also recommend that future studies implement the application of the clinical pathways of non-union fractures and involve the proposed healthcare practitioners, and add more variables that demonstrate the outcomes of clinical pathways.

## 5. Conclusions

In the current study, non-union fracture healing impacted the emotional and physical functioning of patients, particularly because it reduces mobility. Lower-limb non-union fractures significantly reduced the quality of life of patients due to a slew of emotional issues and impaired physical health. In our study, most patients were smokers, had diabetes and were infected with HIV. South African clinicians should consider these factors when planning the treatment of patients’ fractures and use this information to avoid fracture non-union. Although our sample size was small, there may be a link between smoking, diabetes, HIV positivity and non-union fractures. However, we could not test this assumption as this was not the study aim, and the study used a limited sample size of patients with non-union fractures. We recommend the inclusion of social workers, nursing staff experienced in specialized wound care, and psychologists when developing clinical pathways for patients with non-union fractures.

## Figures and Tables

**Table 1 healthcare-11-01810-t001:** The socio-demographic and health status characteristics of patients with non-union lower-limb fractures in state hospitals in the Gauteng province of South Africa.

Characteristics	Category (*n* = 22)	Frequency (%)
Sex	Male	90.9 (*n* = 20)
	Female	9.09 (*n* = 2)
Age	20–29	27.27 (*n* = 6)
	30–39	22.72 (*n* = 5)
	40–49	22.72 (*n* = 5)
	50–59	22.72 (*n* = 5)
	>60	4.54 (*n* = 1)
Marital Status	Married	31.81 (*n* = 7)
	Divorced	9.09 (*n* = 2)
	Widowed	4.54 (*n* = 1)
Type of house	Stand-alone house	63.63 (*n* = 14)
	Flat	18.18 (*n* = 4)
	Shack	18.18 (*n* = 4)
Employment status	None	68.18 (*n* = 15)
	Student	13.63 (*n* = 3)
	Pensioner	4.54 (*n* = 1)
Comorbidity	Hypertension	18.18 (*n* = 4)
	Smoking	86.36 (*n* = 19)
	HIV+	36.36 (*n* = 8)
Duration of non-union fractures	Less than a year	9.1 (*n* = 2)
	More than a year	72.72 (*n* = 16)
	More than two years	18.18 (*n* = 4)

**Table 2 healthcare-11-01810-t002:** The clinical pathways of patients with lower-limb non-union fractures.

Variable	% (*n*)
Diagnosis	
Distal femur fracture	22.7 (*n* = 5)
Proximal tibia fracture	13.6 (*n* = 3)
Proximal tibia/fibula fracture	4.6 (*n* = 1)
Midshaft tibia fracture	18.2 (*n* = 4)
Distal tibia fracture	22.7 (*n* = 5)
Distal tibia/fibula fracture	4.6 (*n* = 1)
Medial malleolus fracture	13.6 (*n* = 3)
International classification of function, diseases and disabilities	
Impairment	100 (*n* = 22)
Activity limitation	100 (*n* = 22)
Participation restriction	100 (*n* = 22)
Re-admission within six months	
Yes	54.6 (*n* = 12)
No	45.4 (*n* = 10)

**Table 3 healthcare-11-01810-t003:** Quality of life domains according to the lower-limb fracture location.

Domains	Diagnosis	Number of Observations	Rank Sum	K–Wallis Test
Physical Function	Femur	5	66.0	0.59
	Tibia	12	146.5	
	Tibia/fibula	2	16.0	
	Malleolus	3	2.5	
Mental Health	Femur	5	57.5	1.00
	Tibia	12	138.0	
	Tibia/fibula	2	23.0	
	Malleolus	3	34.5	
Vitality	Femur	5	63.5	0.96
	Tibia	12	132.5	
	Tibia/fibula	2	21.0	
	Malleolus	3	36.0	
Emotional Role	Femur	5	68.0	0.03
	Tibia	12	166.0	
	Tibia/fibula	2	7.0	
	Malleolus	3	12.0	
Social Functioning	Femur	5	43.5	0.61
	Tibia	12	156.0	
	Tibia/fibula	2	23.5	
	Malleolus	3	30.0	
Pain	Femur	5	62.0	0.57
	Tibia	12	148.0	
	Tibia/fibula	2	23.0	
	Malleolus	3	20.0	
General Health	Femur	5	54.5	0.73
	Tibia	12	153.5	
	Tibia/fibula	2	18.5	
	Malleolus	3	26.5	
Physical Health	Femur	5	57.5	0.00
	Tibia	12	138.0	
	Tibia/fibula	2	23.0	
	Malleolus	3	34.5	

**Table 4 healthcare-11-01810-t004:** Results of the quality of life assessment using SF-36 scoring.

Domains Median (IQR)	Femur (*n* = 5)	Tibia (*n* = 12)	Tibia/Fibula (*n* = 2)	Malleolus (*n* = 3)
PCS Domains				
Physical Function	15 (14, 15)	14.5 (13.5, 16.5)	13 (11, 15)	14 (11, 15)
Physical Role	4 (4, 4)	4 (4, 4)	4 (4, 4)	4 (4, 4)
Body Pain	9 (8, 9)	8 (8, 9)	8.5 (7, 10)	10 (8, 11)
General Health	17 (16, 17)	17 (16, 18)	16.5 (16, 17)	17 (15, 17)
MCS Domains				
Mental Health	17 (16, 20)	19 (17, 20.5)	18 (18, 18)	18 (16, 18)
Social Function	5 (4, 5)	4.5 (4, 5)	4 (3, 5)	5 (3, 7)
Emotional Role	3 (3, 3)	3 (3, 3)	3 (3, 3)	3 (3, 3)
Vitality	13 (12, 14)	11.5 (11, 13.5)	14 (13, 15)	10 (7, 14)

## Data Availability

Data for this study are available upon request.

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
