# Peer review of "Factors Informing the Development of a Clinical Pathway and Patients’ Quality of Life after a Non-Union Fracture of the Lower Limb"

_healthcare, 2023, doi:10.3390/healthcare11121810_

Round 1

Reviewer 1 Report

This prospective study aimed to investigate patients' clinical pathways and quality of life with non-union fractures in their lower limbs. The study collected data from hospital records, utilizing a questionnaire to track patients' follow-up frequency, daily activities, and outcomes at six months. The initial quality of life was assessed using the short form-36 questionnaire. The study found that patients with lower limb non-union fractures experienced impairments, limited activity, and participation restrictions. In addition, these fractures significantly impacted both emotional and physical health, highlighting the need for a more comprehensive approach to patient care. The authors are commended for their hard work conducting a complex, time-consuming investigation. Their methodology is well-crafted and designed. However, to improve the readability and accuracy of their work, the following comments should be considered.
1) The authors should present questionnaire forms employed in this study in the appendix section since these forms are a focal point of the investigation.
2) The authors need to provide more examples of how clinical pathways may play an important role in the treatment process; most of the presented descriptions were general. This will endorse the scientific question investigators are trying to answer via this work.

3) Figure N1: is the only figure in this work, and it looks like that copied as is it from the used software, a raw figure. Here the authors have to spend more time presenting this figure better. The whole spectrum of answers should be present, and here they can use the BOXCHAT or Violin plot.

4) There is a great set of published papers in the healthcare journal  (https://www.mdpi.com/journal/healthcare) that may endorse the list of references of the proposed paper; the reviewer encourages the authors to cite those papers, which may increase the visibility of the proposed paper within and out the frame of the publisher.   

Reviewer 2 Report

TThe study is interesting because it focuses on troublesome non-union fractures of the lower limb, but contains various statistical insufficiencies.

The authors said as a result that HIV-infected patients more likely to cause non-union fractures, but the HIV infection rate in healthy subjects as a control is unknown. It is also impossible to determine whether HIV-infected patients are more likely to cause non-union fractures or whether non-union fractures are more common in HIV-infected patients simply because they are more likely to have fractures in the lower limb. And also no information is provided to determine whether the fracture site is simply prone to fracture or prone to non-union fracture if it once fractured.

The results of the Kruskas-Wallis test for physical and mental function assessment shown in Table 3 seem to be statistically meaningful, but there is a high possibility of various biases in the test results of a relatively small number of cases, which were conducted without adjusting for large variations in age and other factors, and though It seems reasonable that fractures of minor parts such as malleolus had significantly higher role emotional scores and physical health scores, it seems that the impact of obtained information is small because it was rather expected.

In table 3, the rightest column is not chi-square values but p values of K-W test.

Figure 1 is not a particularly easy-to-read diagram, and it would be better if the numerical values ​​were properly included in Table 3 with SDs. Since nonparametric tests are used, it may be better to show the medians instead of the means.

Reviewer 3 Report

Congrats for your paper!

I think that the investigation is complete, however the article could improve with more information about the physical therapy treatment in non-union fractures.

Maybe, the future interventions should include more variables.

Round 2

Reviewer 2 Report

It seems that the manuscript has improved sufficiently to a publishable level.